# Prognostic Factor of Lower Limb Amputation among Diabetic Foot Ulcer Patients in North-East Peninsular Malaysia

**DOI:** 10.3390/ijerph192114212

**Published:** 2022-10-31

**Authors:** Anas Rosedi, Suhaily Mohd Hairon, Noor Hashimah Abdullah, Nor Azwany Yaacob

**Affiliations:** 1Department of Community Medicine, School of Medical Sciences, Health Campus, Universiti Sains Malaysia, Kubang Kerian 16150, Kelantan, Malaysia; 2Non-Communicable Disease Unit, Disease Control Division, Kelantan State Health Department, Ministry of Health Malaysia, Jalan Mahmood, Kota Bharu 15200, Kelantan, Malaysia

**Keywords:** prognostic factor, Cox model, regression, amputation, diabetic foot, Malaysia

## Abstract

Lower limb amputation (LLA) is a common complication of diabetic foot ulcer (DFU), which can lead to a higher 5-year mortality rate compared to all cancers combined. This study aimed to determine the prognostic factors of LLA among DFU patients in Kelantan from 2014 to 2018. A population-based study was conducted using secondary data obtained from the National Diabetic Registry (NDR). There were 362 cases that fulfilled the study criteria and were further analysed. The prognostic factors were determined by Multiple Cox Proportional Hazards Regression. There were 66 (18.2%) DFU patients who underwent LLA in this study, while 296 (81.8%) were censored. The results revealed that the factor leading to a higher risk of LLA was abnormal HDL-cholesterol levels (Adj. HR 2.18; 95% CI: 1.21, 3.92). Factors that led to a lower risk of LLA include DFU in patients aged 60 or more (Adj. HR 0.48; 95% CI: 0.27, 0.89) and obesity (Adj. HR 0.45; 95% CI: 0.22, 0.89). In conclusion, our model showed that abnormal HDL cholesterol was associated with a 2 times higher risk of LLA when adjusted for age and BMI. Any paradoxical phenomena should be addressed carefully to avoid wrong clinical decision making that can harm the patient.

## 1. Introduction

The global trend of Diabetes Mellitus (DM) is worrying. The total number of people with DM is projected to continue increasing. By 2030, the overall incidence of DM is expected to rise to 578 million (10.2%), and by 2045, it will rise to 700 million (10.9%). DM also contributes to global NCD deaths, having been identified as the fourth most common cause of NCD death at around 1.6 million [1]. DFU is one of the commonest diabetic complications. It is characterised as the ulceration or destruction of foot tissues caused by neuropathy or peripheral arterial disease (PAD) among people with DM [2].

The International Diabetic Federation reported the global prevalence of DFU to be 6.3%, but it could range up to 14% [3,4,5]. The prevalence of DFU in Malaysia has generally ranged from 5–10%, but in a study conducted in 2018, the prevalence of DFU reached as high as 42% among diabetic patients in Kuala Lumpur [6,7]. D. G. Armstrong et al. found that the most significant risk factor for a DFU was a healed DFU [8]. DFU was the leading cause of LLA, and 85% of poorly healed DFU cases ended in LLA because the neuropathy and PAD persisted and readily caused another ulcer if proper foot care was not established [8,9]. Furthermore, Zhang et al. reported that 25% of diabetic patients would develop diabetic foot ulcers in their lifetime [4].

In their study in 2007, D. G. Armstrong et al. reemphasised the significant impact of DFU and LLA on the mortality rate. Their 5-year mortality data revealed that overall LLA led to a higher 5-year mortality rate compared to all cancers combined. The 5-year mortality with major LLA was 56.6%, and minor LLA led to 46.2% mortality. Hence, they concluded that both DFU and LLA were independent risk factors for premature mortality [10]. 

The increasing number of undiagnosed DM cases is a worrying trend, as it is expected that more patients will have already established complications, such as DFU, upon diagnosis. The rising prevalence of diabetes, especially undiagnosed diabetes, will lead to an increasing number of diabetic feet at risk of amputation. By identifying the prognostic factors of LLA, a better strategy can be formulated to manage diabetic foot ulcers and prevent LLA from taking place.

## 2. Materials and Methods

Study design: This study was a retrospective cohort study that was conducted from 1 October 2020 to 31 May 2021. The sampling frame was all DFU patients in Kelantan registered in the NDR from 2014 to 2018 who fulfilled the study criteria. Patients with DFU diagnosed from 1 January 2014 to 31 December 2018 who were registered in NDR and were treated in the Government Health Facility in Kelantan were included in this study. Patients with an error in the diagnosis date for DFU/LLA or patients with >30% incomplete data were excluded from the study sample. 

Sample Size: The sample size to determine the prognostic factors of LLA among DFU patients was calculated using two median time options in PS Software: Power and Sample Size Calculation version 3.1.6. The two median times entered were 14 months and 20 months [11]. The accrual time was 60 months, and the follow-up time was 12 months, which led to a calculated sample size of 360. After adding a 10% dropout rate, the total sample size was 400 patients. 

Data collection: This study was conducted using secondary data, whereby the data were extracted from the National Diabetic Registry (NDR), which consists of registry datasets and clinical audit data compiled by the Kelantan State Health Department. The NDR is an innovation of the Ministry of Health (MOH) for diabetic surveillance. It is a web-based registry that can enable a systematic approach to collecting data and monitoring the quality of care and type 2 DM patients managed in the MOH facility [12]. Names and identification numbers were made anonymous and coded into numbers for analysis purposes. Proforma was used as a checklist of data required to be analysed in this study while ensuring confidentiality. The independent variables were extracted from the NDR and DM Audit Data. The data from DM Audit Data were obtained at or after the diagnosis of DFU was made and before amputation took place to determine the hazard relationship. Statistical Analysis: These data were downloaded into Microsoft Excel and later exported to SPSS Software version 26 for further analysis. Descriptive statistics were used to summarise the characteristics of each sample. Finally, the categorical data were presented as frequency (*n*) and proportion (%). Normality tests were conducted using SPSS and manual calculations. Given that non-Malay cases represented very small numbers in the sample compared to Malay cases, ethnicities other than Malay, such as Chinese, Indian, Siamese, and others, were recoded into one category: “non-Malay”. HDL cholesterol was categorised as normal and abnormal based on the cut-off point stated in the Clinical Practice Guideline on Management of Dyslipidaemia 2017, where less than 1.0 mmol for males and less than 1.2 mmol/L for females are considered normal [13]. This study used the survival analysis method. The event of interest was LLA. Hence, the duration from the diagnosis of DFU to the date when LLA occurred was called amputation-free time. This cohort was followed up from the date of diagnosis until the date of the event of interest, LLA. Patients who did not experience amputation, who were lost to follow-up, or who died before the event took place until 31 December 2019 were categorised as censored data. Univariable (Simple Cox Proportional Hazards Regression) and multivariable analyses (Multiple Cox Proportional Hazards Regression) were performed to determine the prognostic factors of LLA among the independent variables. Univariable analyses were used to determine which factors are significant at the univariate level, where the variables of choice to proceed with Multiple Cox Proportional Hazards Regression were those with a *p*-value less than 0.25 or those that were clinically significant. The variables of choice that were obtained from Simple Cox Proportional Hazards Regression were analysed further using Multiple Cox Proportional Hazards Regression. In this multivariate analysis, the variables of choice were those with *p*-value < 0.25 or those that were clinically significant and underwent variable selection using Backward and Forward LR methods. The best model selected was checked further for any two-way interactions. Once no interaction was identified, the preliminary main effect model was obtained. There were two assumptions to be fulfilled in the proportional hazards analysis. The first assumption was that the hazard function is constant over time. This assumption would compare the effect of two or more factors that were superior or were more hazardous to the outcome over time. The second assumption was that the hazard is proportional over time. This assumption is expressed as the hazard ratio, which is the ratio between two expected hazards.

## 3. Results

There were 598 DFU cases registered in the NDR from 2014 to 2018. After 236 cases were removed because they were duplicates or fulfilled the exclusion criteria, 362 remained for further analysis (Figure 1).

### 3.1. Descriptive Statistics of DFU Patients

The description of patients’ profiles with DFU in Kelantan from 2014 to 2018 is shown in Table 1. Most of the patients were of Malay ethnicity (96.7%), with a mean (SD) age of 59.0 (10.0) years old, and most were female (61.6%), obese according to BMI (55.9%), and non-smokers (92.8%). The ratio of males to females in this study was 1:1.6.

In terms of comorbidities, most DFU patients in this study had hypertension (81.1%) and dyslipidaemia (77.6%). On the other hand, only 26.9% had nephropathy, and 12.1% had ischaemic heart disease. For the biochemical parameters, 87.5% of DFU patients in this study had Hba1c levels of 6.5% or more, but only 55.5% had triglyceride levels of more than 1.7. In addition, 75.3% had an LDL-cholesterol level of more than 2.6, but only 37.3% had an abnormal HDL-cholesterol level. From the treatment perspective, most patients were on Statins (76.2%), Insulin (63.5%), and Metformin (65.7%), whereas only 29.6% were on Sulphonylurea.

### 3.2. Univariate and Multivariate Analysis

The univariate analysis results revealed that patients who were on Sulphonylurea (HR 0.50; 95% CI: 0.27, 0.94; *p* = 0.019) showed a statistically significant protective hazard ratio. Older age at diagnosis of DFU showed a significantly lower hazard ratio (HR 0.54; 95% CI: 0.33, 0.89; *p* = 0.016). Table 2 showed the overall result of univariate analysis.

The results obtained from the univariate analysis were further analysed using multivariate analysis, which was Multiple Cox Proportional Hazards Regression. Variables with a *p*-value less than 0.25 were included in the variable selection. The preliminary final model was further analysed to ensure that there were no interactions and that all assumptions were met.

The final model for the LLA prognostic factors is shown in Table 3. After adjusting for the other variables, patients with DFU aged 60 or more had a 52% lower risk of LLA compared to DFU patients aged less than 60 (Adj. HR 0.48; 95% CI: 0.27, 0.89). Patients with DFU who were obese had a 55% lower risk of LLA than patients with DFU who had a normal BMI (Adj. HR 0.45; 95% CI: 0.22, 0.89). Patients with abnormal HDL cholesterol had a 2.18 times higher risk of LLA than patients with normal HDL-cholesterol levels (Adj. HR 2.18; 95% CI: 1.21, 3.92).

## 4. Discussion

Data from the National Diabetic Registry, a well-established public health monitoring system, were used in this retrospective cohort analysis. This registry’s data include all DFU cases and their amputation status, including the date of diagnosis, and are maintained regularly. However, for this review, only DFU patients who were followed up by the Kelantan Government Health Facility and were diagnosed between 1 January 2014 and 31 December 2018 were chosen. As a result, the data used in this study are accurate for and descriptive of Kelantan’s DFU patients.

Since the study’s emphasis was on the prognostic factors of LLA, survival analysis was the chosen method. These analyses considered the unmeasured absolute amputation-free time among patients who had not had their limbs amputated at the end of the analysis. Some researchers have studied all types of amputation, major amputation, and survival after amputation [11,14,15,16]. Therefore, LLA of all types was valid for use as the event of interest in this study. However, our model includes two results that seem contradictory to the current knowledge, where older age and obese BMI appear to be protective against LLA.

Literature searches mostly reveal that ageing can lead to a higher risk of LLA [15,16]. Established evidence suggests that older age leads to delayed wound healing. Ageing has been associated with delayed inflammation, angiogenesis, and epithelialisation, leading to delayed wound healing [17]. The first paradox in this study was that older age appeared to be protective against LLA. For healing to occur, biological capacity is not the only consideration, as compliance with follow-up for wound treatment is also crucial in determining the outcome of DFU. Older people have been shown in various literature studies to have a better compliance rate with follow-up and medications [18,19,20]. Moreover, in Malaysia, wound clinics have been established in most primary care clinics that these older patients can easily access. These clinics provide standardised wound care according to the national manual at an affordable fee [21]. Further studies can be conducted to prove the confounding effect of clinical follow-up on the outcome of DFU. 

The second paradox found in this study was that obesity appeared to be protective against LLA. This phenomenon is popularly termed the obesity paradox, where obesity appears to have a positive effect on health outcomes. This observation has appeared in multiple observational studies involving study populations with existing chronic diseases [22,23]. However, the low-BMI status in chronic disease patients cannot be compared to having a low BMI in the general population. Having a low BMI for patients with existing chronic diseases can mean that the disease is poorly controlled, and hence, they are more likely to develop poor health outcomes, such as amputation [24]. As shown in this study, most had uncontrolled diabetes, where those with HbA1c of 6.5% or more were more likely to undergo LLA (87.5%) compared to those with HbA1c less than 6.5% (12.5%). Hence, this could be the explanation for the “obesity paradox” seen in this study.

HDL plays a key role in reverse cholesterol transfer (RCT). RCT is a process in which excess cholesterol is extracted from peripheral vessels and transferred back to the liver for disposal [25]. On the other hand, HDL has several other biological properties that help it defend against CVD. The antioxidative properties of HDL particles in the circulation, especially small, dense, protein-rich HDL3, can protect LDL from oxidative damage by free radicals in the arterial intima [26]. In addition, HDL can exert a direct cytoprotective effect on endothelial cells by inhibiting the suicide pathway that leads to endothelial cell apoptosis by lowering cysteine protease P32 (CPP32)-like protease activity. As a result, HDL protects against “injury”, as defined in the atherogenesis “response-to-injury” hypothesis [27,28]. The protective hazard ratio in this study is consistent with other research in Japan, in which patients with lower HDL cholesterol were reported to have an association with a higher risk of minor and major amputations [29]. Kaneko et al. also reported that HDL cholesterol was associated with a 33% lower risk of LLA, but the result was not statistically significant [30]. This study complements these findings by proving that normal HDL cholesterol significantly lowered the LLA risk, even after controlling for age and BMI status. 

This study used data from the NDR, a registry of diabetic patients who were followed up by government health clinics in Malaysia, specifically in Kelantan. The majority of DM patients in Malaysia are followed up by government health clinics. So, the NDR is a reliable tool for use as a surveillance system for diabetic patients, their complications, and their comorbidities in Malaysia. The registry is continuously maintained and updated by a well-trained staff in every health clinic in Malaysia. The Kelantan State Health Department conducts annual training in Kelantan to ensure that every clinic has at least one health worker capable of using the system and regularly updating necessary information. Every district in Kelantan also has its own medical and health officer in charge of the diabetic program. They supervise and ensure that the system is well-maintained at the district level. However, like other secondary data, the data quality was not free from uncertainty, as errors may still arise when staff key in their clinic data. Another limitation is that this study was unable to include diabetic patients who had their follow-ups in private clinics, as they are not registered in the NDR. However, the number is small compared to the 74.3% of diabetic patients in Malaysia who seek treatment in government health clinics [12].

There are other variables that were significant in other studies but were found to be non-significant in this study, such as smoking, the duration of diabetes, gender, comorbidities, and treatments such as Insulin and Oral Anti-Diabetic Medication. The non-significance could be due to the result of confounding effects, and perhaps a larger sample size is required.

## 5. Conclusions

The prognostic factor of LLA found in this study was abnormal HDL cholesterol, which was associated with a 2 times higher risk of LLA when adjusted for age and BMI. This prognostic factor can be used by healthcare professionals to give more attention to high-risk cases and to suggest appropriate action in their plan and strategy to prevent LLA among DFU patients. The prevention of DFU will not only improve the patient’s quality of life but also help in increasing the patient’s survival. Any paradoxical findings should be doubted and carefully examined to ensure that they do not lead to wrong clinical decisions that can harm patients.

## Figures and Tables

**Figure 1 ijerph-19-14212-f001:**
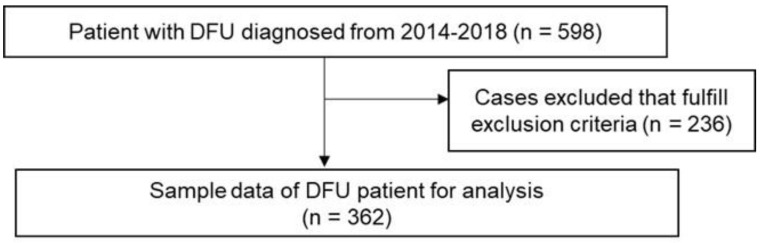
Flow Chart of Sample.

**Table 1 ijerph-19-14212-t001:** Characteristics of patients with DFU in Kelantan from 2014 to 2018.

Characteristic	*n* (%)	Censored	Amputation
		*n* (%)	*n* (%)
Age at diagnosis of DFU (years)	59.0 (10.0) *		
Age at diagnosis of DFU (years)			
Less than 60	175 (48.34)	134 (76.6)	41 (23.4)
60 or more	187 (51.66)	162 (86.6)	25 (13.4)
Sex			
Male	139 (38.4)	116 (83.5)	23 (16.5)
Female	223 (61.6)	180 (80.7)	43 (19.3)
Ethnicity			
Malay	350 (96.7)	285 (81.4)	65 (18.6)
Non-Malay	12 (3.3)	11 (91.7)	1 (8.3)
Duration of DM at diagnosis DFU (years)			
Less than 10	211 (58.3)	171 (81.0)	40 (19.0)
10 or more	151 (41.7)	125 (82.8)	26 (17.2)
Smoking status			
No	296 (92.8)	242 (81.8)	54 (18.2)
Yes	23 (7.2)	20 (87.0)	3 (13.0)
Body Mass Index			
Normal	74 (23.0)	58 (78.4)	16 (21.6)
Underweight	14 (4.4)	8 (57.1)	6 (42.9)
Overweight	54 (16.8)	45 (83.3)	9 (16.7)
Obese	180 (55.9)	158 (87.8)	22 (12.2)
Hba1c (%)			
Less than 6.5	39 (12.5)	33 (84.5)	6 (15.5)
6.5 or more	272 (87.5)	222 (81.6)	50 (18.4)
Triglyceride (mmol/L)			
Less than 1.7	130 (44.5)	107 (82.3)	23(17.7)
1.7 or more	162 (55.5)	131 (80.9)	31(19.1)
HDL-Cholesterol			
Normal	170 (62.7)	143 (84.1)	27 (15.9)
Abnormal	101 (37.3)	75 (74.3)	26 (25.7)
LDL-Cholesterol (mmol/L)			
Less than 2.6	66 (24.7)	54 (81.8)	12 (18.2)
2.6 or more	201 (75.3)	161 (80.1)	40 (19.9)
Ischaemic Heart Disease			
No	299 (87.9)	246 (82.3)	53 (17.7)
Yes	41 (12.1)	30 (73.2)	11 (26.8)
Nephropathy			
No	253 (73.1)	209 (82.6)	44 (17.4)
Yes	93 (26.9)	73 (78.5)	20 (21.5)
Hypertension			
No	67 (18.9)	59 (88.1)	8 (11.9)
Yes	287 (81.1)	231 (80.5)	56 (19.5)
Dyslipidaemia			
No	79 (22.4)	67 (84.8)	12 (15.2)
Yes	273 (77.6)	220 (80.6)	53 (19.4)
Metformin			
No	124 (34.3)	99 (79.8)	25 (20.2)
Yes	238 (65.7)	197 (82.8)	41 (17.2)
Sulphonylurea			
No	255 (70.4)	201 (78.8)	54 (21.2)
Yes	107 (29.6)	95 (88.8)	12 (11.2)
Insulin			
No	132 (36.5)	114 (86.4)	18 (13.6)
Yes	230 (63.5)	182 (79.1)	48 (20.9)
Statin			
No	86 (23.8)	72 (83.7)	14 (16.3)
Yes	276 (76.2)	224 (81.2)	52 (18.8)

* Mean (SD).

**Table 2 ijerph-19-14212-t002:** Univariate analysis using Simple Cox Regression.

Variables	Regression Coefficient (β)	Crude Hazard Ratio (95% CI)	Wald Stats	*p*-Value
Sex				
Male		1		
Female	0.18	1.20 (0.72, 2.00)	0.49	0.485
Age at diagnosis of DFU (years)				
Less than 60		1		
60 or more	−0.61	0.54 (0.33, 0.89)	5.79	0.016
Ethnic				
Malay		1		
Non-Malay	−0.85	0.43 (0.06, 3.08)	0.71	0.398
Duration of DM at diagnosis of DFU				
Less than 10		1		
10 years and more	−0.06	0.94 (0.57, 1.54)	0.07	0.798
BMI				
Normal		1		
Underweight	−0.69	1.98 (0.78, 5.07)	2.05	0.153
Overweight	−0.27	0.76 (0.34, 1.72)	0.43	0.512
Obese	−0.63	0.53 (0.28, 1.01)	3.69	0.055
Hba1c (%)				
Less than 6.5		1		
6.5 or more	0.19	1.21 (0.52, 2.82)	0.19	0.661
Triglyceride (mmol/L)				
Less than 1.7		1		
1.7 or more	0.06	1.06 (0.62, 1.82)	0.05	0.828
HDL-Cholesterol				
Normal		1		
Abnormal	0.52	1.69 (0.99, 2.90)	3.64	0.056
LDL-Cholesterol (mmol/L)				
Less than 2.6		1		
2.6 or more	0.10	1.10 (0.58, 2.09)	0.08	0.773
Smoking Status				
No		1		
Yes	−0.32	0.72(0.23, 2.32)	0.30	0.586
Ischaemic Heart Disease				
Absent		1		
Present	0.44	1.55 (0.81, 2.96)	1.74	0.187
Nephropathy				
Absent		1		
Present	0.24	1.28 (0.75, 2.17)	0.81	0.368
Hypertension				
Absent		1		
Present	0.53	1.71 (0.81, 3.58)	1.99	0.158
Dyslipidaemia				
Absent		1		
Present	0.24	1.27 (0.68, 2.37)	0.55	0.457
Metformin				
No		1		
Yes	−0.17	0.84 (0.51, 1.39)	0.46	0.499
Sulphonylurea				
No		1		
Yes	−0.69	0.50 (0.27, 0.94)	4.70	0.030
Insulin				
No		1		
Yes	0.47	1.59 (0.93, 2.74)	2.83	0.093
Statin				
No		1		
Yes	0.13	1.13 (0.63, 2.05)	0.173	0.677

β, Regression coefficient; HR, hazard ratio; CI, confidence interval.

**Table 3 ijerph-19-14212-t003:** The Final Model for Prognostic Factors of LLA among DFU Patients.

Variables	Regression Coefficient (β)	Adjusted Hazard Ratio (95% CI)	Wald Stats	*p*-Value
Age at diagnosis of DFU (years)				
Less than 60		1		
60 or more	−0.73	0.48 (0.27, 0.89)	5.52	0.019
BMI (kg/m^2^)				
Normal		1		
Underweight	0.77	2.17 (0.78, 6.07)	2.17	0.140
Overweight	−0.39	0.68 (0.28, 1.62)	0.77	0.381
Obese	−0.81	0.45 (0.22, 0.89)	5.22	0.039
HDL-Cholesterol				
Normal		1		
Abnormal	0.78	2.18 (1.21, 3.92)	4.55	0.009

Forward stepwise Cox proportion hazards regression model applied; log-minus-log plot and hazard function plot were applied to check the model assumption; regression equation: h(t) = [h_0_(t)] e ^[(−0.73 × Age) + (−0.81 × BMI Obese)) + (0.78 × HDL-Cholesterol)^.

## Data Availability

The data from the National Diabetic Registry used to support this study have not been made available because of the privacy and confidentiality of the data.

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
