# Peer review of "Prognostic Factor of Lower Limb Amputation among Diabetic Foot Ulcer Patients in North-East Peninsular Malaysia"

_ijerph, 2022, doi:10.3390/ijerph192114212_

Round 1
Reviewer 1 Report
Introduction: The rationale of the study is not clear. This section should be improved. It appears more like a list of studies and findings.
Methods: I can not understand the measures you collected and the reason you selected them. The auhtors only wrote ' The independent variable was broadly classified into 4 categories, sociodemographic factor, biochemical factor, and treatment factor'. You should furnish detailed descriptions of these parameters. E.g. What Biochemical factors are you including?
Discussion: This section needs to be improved because it is unclear. You should compare your findings with the current knowledge and justifies the possible differences. The article appears more like a review article that an original article.
Author Response
Greetings,
First of all, thank you for your insightful comments and advise, it has been an honor to receive reply from IJERPH reviewer. We have summarized my response on your comments as shown in the points below:
Point 1: Introduction: The rationale of the study is not clear. This section should be improved. It appears more like a list of studies and findings.
Response 1: The introduction section and rationale of the study was revised. In summary, we would like to point out that the increasing prevalence of diabetes also leads to the increasing number of DFU, which put them at risk of LLA and premature death.
Point 2: Methods: I can not understand the measures you collected and the reason you selected them. The auhtors only wrote ' The independent variable was broadly classified into 4 categories, sociodemographic factor, biochemical factor, and treatment factor'. You should furnish detailed descriptions of these parameters. E.g. What Biochemical factors are you including?
Response 2: We agreed and have omitted the phrases as they can be misleading because the table has already included all without partitioning it into the section.
Point 3: Discussion: This section needs to be improved because it is unclear. You should compare your findings with the current knowledge and justifies the possible differences. The article appears more like a review article that an original article.
Response 3: Thank you for pointing this out, we have re-done the discussion to avoid making it looks like a review article.
In addition to the above comments, all spelling and grammatical errors pointed out have been corrected.
Thank you again for your wisdom and insightful comments. We hope that our explanations and responses could allow our manuscript to be accepted for publication in IJERPH
Sincerely,
Anas Rosedi
Reviewer 2 Report
This is a quality manuscript, which requires text editing, better structure, and clarification of your ideas. I have summarised my recommendations below:
Presentation:
I suggest that the authors fix spaces between the sentences. Sometimes, they have double space, sometimes no space.
Capitalise T in the second sentence of the abstract.
Include the full stop in the sentence staring with: As a result, the data used in this study was accurate and descriptive of Kelantan's DFU patients
Full stop after: However, local research might need to be done to assure the possible explanation of 200 this study's finding where older age appeared protective of LLA
Please reword - was chosen as the chosen method
Discussion:
Re-state the aim of your study in the beginning of this section.
Your discussion is difficult to read as it is unclear what was found in your study and studies that you compared with. For example:
Obesity was associated with many adverse health effects and had established itself as a significant risk factor with many comorbidities. Is this your finding? If not, please support your claim with various other studies.
Another example:
But this study, like other research, reported obesity as becoming a protective factor against certain diseases—this condition was called the obesity paradox. Like other research requires at least 3 citations
Also, ‘This study and their study confirm’ lacks clarity
And many other sentences in Discussion lack clarity
In Discussion, please make shorter paragraphs to improve the clarity of their content
Your argument was lost, and it is unclear, how you have explained the paradox, perhaps you better summarise your ideas in the beginning of the Discussion.
Limitations:
Study limitations should be identified and discussed
Tables:
Please left align the content of the first column in all tables
Check the numbers of the included tables
References:
Please follow the Journal referencing style. You used both numbered and the Author-date styles.
Author Response
Greetings,
First of all, we would like to thank you for the compliment saying our manuscripts is a quality manuscript, it has been an honor to be guided by respected reviewer from IJERPH. We have written the response to your comments as shown below.
Point 1:
Presentation:
I suggest that the authors fix spaces between the sentences. Sometimes, they have double space, sometimes no space.
Capitalise T in the second sentence of the abstract.
Include the full stop in the sentence staring with: As a result, the data used in this study was accurate and descriptive of Kelantan's DFU patients
Full stop after: However, local research might need to be done to assure the possible explanation of 200 this study's finding where older age appeared protective of LLA
Please reword - was chosen as the chosen method
Response 1:
This was an oversight, and we apologize on the matter. We have fixed the error that was pointed out and reword accordingly.
Point 2:
Discussion:
Re-state the aim of your study in the beginning of this section.
Your discussion is difficult to read as it is unclear what was found in your study and studies that you compared with.
For example:
Obesity was associated with many adverse health effects and had established itself as a significant risk factor with many comorbidities. Is this your finding? If not, please support your claim with various other studies.
Another example:
But this study, like other research, reported obesity as becoming a protective factor against certain diseases—this condition was called the obesity paradox. Like other research requires at least 3 citations.
Also, ‘This study and their study confirm’ lacks clarity
And many other sentences in Discussion lack clarity
In Discussion, please make shorter paragraphs to improve the clarity of their content
Your argument was lost, and it is unclear, how you have explained the paradox, perhaps you better summarise your ideas in the beginning of the Discussion.
Response 2: Thank you for sharing your wisdom and suggestions here. We have already re-done the discussion by first justifying the study design and then followed by explanation of paradox finding. we have also re-done the discussion in hope that it provides more clarity as pointed out.
Point 3:
Limitations:
Study limitations should be identified and discussed
Response 3: We have addressed the limitations in the last paragraph of the discussion. Kindly refer to line 332 to 341.
Point 4:
Tables:
Please left align the content of the first column in all tables
Check the numbers of the included tables
Response 4: We have already done the realignment, actually we are unsure why our manuscript table's alignment went haywire after being uploaded initially, hope that this corrected manuscript will not be the same.
We also went through the number and corrected it accordingly.
Point 5:
Please follow the Journal referencing style. You used both numbered and the Author-date styles.
Response 5: This is another oversight from our side, we have changed it to numbered citations. Please accept our apology and thank you for pointing this out.
We would like to thank you again for your wisdom and insightful comments. We hope that our explanations and responses could allow our manuscript to be accepted for publication in IJERPH
Sincerely
Anas Rosedi
Reviewer 3 Report
The current study is an excellent try to investigate the possible risk factors which could cause DFU and LLA.
But, in that kind of study, there are too many limitations besides the small number of participating patients.
The main limitations are: the duration of DM is not known, maybe the degree of neuropathy, peripheral artery disease, ischaemic heart disease, stroke, poor glycemic control, diabetic foot care, smoking, lipidemic status, hypertension control, management of all comorbidities and BMI during the entire course of DM.
Moreover, the values of laboratory measurements had not been reported.
I think that the methodology is appropriate. The manuscript is well written, but the discussion/conclusions are not acceptable.
Overall, data could be of interest, but I cannot approve the publication of the current study because the authors suggest a wrong message for clinicians.
Author Response
Greetings,
First of all, it has been an honor on our side to receive reply from IJERPH reviewer. Thank you for your kind, insightful comments and suggestions. We have listed our responses to the comments as listed below.
Point 1: The current study is an excellent try to investigate the possible risk factors which could cause DFU and LLA.
But, in that kind of study, there are too many limitations besides the small number of participating patients.
Response 1: We agreed that the relatively small number is indeed a limitation because we are using secondary data from the Malaysia National Diabetic Registry (NDR).
Point 2: The main limitations are: the duration of DM is not known, maybe the degree of neuropathy, peripheral artery disease, ischaemic heart disease, stroke, poor glycemic control, diabetic foot care, smoking, lipidemic status, hypertension control, management of all comorbidities and BMI during the entire course of DM.
Response 2: The time to event in this study is from the date of diagnosis of DFU to the date of amputation recorded in the registry. We agreed that the duration of DM is hard to determine due to long latent period, but it should have minimal effect on this study because the DFU and amputation is a clinical diagnosis and both dates are clearly stated in the registry.
Point 3: Moreover, the values of laboratory measurements had not been reported. I think that the methodology is appropriate. The manuscript is well written, but the discussion/conclusions are not acceptable.
Response 3: All laboratory variables in this study have their respective cutoff point and unit used except for HDL, this is because the cutoff points differ between male and female (1.0mmol/L for male and 1.2mmol/L for female), hence we categorized it into normal and abnormal accordingly. We have also added this explanation in the method section. About the discussions and conclusions, we have re-done them to provide more clarity on our stands.
Point 4: Overall, data could be of interest, but I cannot approve the publication of the current study because the authors suggest a wrong message for clinicians.
Response 4: We apologize for the lack of clarity on our stands in the discussion and conclusion. We have re-done the discussion and conclusion to provide more clarity on our stands and to ensure that the right message is conveyed to the clinicians who referred to this manuscript. Our manuscript title is also changed to prevent further misunderstanding.
We would like to thank you again for your wisdom and insightful comments. We hope that our explanations and responses could allow our manuscript to be accepted for publication in IJERPH
Sincerely,
Anas Rosedi
Round 2
Reviewer 2 Report
Dear Authors
Thank you for revising your manuscript. I was satisfied with the nature and the scope of your revisions; and I recommended publication.
I am waiting forwards to see your article in press.
Kind regards,